# Survey on the Mental Health of Dispensing Pharmacists in the Auvergne-Rhône-Alpes Region (France)

**DOI:** 10.3390/ijerph20216988

**Published:** 2023-10-28

**Authors:** Bernard Massoubre, Tristan Gabriel-Segard, Florence Durupt, Anne-Sophie Malachane, Noémie Anglard, Théophile Tiffet, Catherine Massoubre

**Affiliations:** 1Institute of Pharmaceutical and Biological Sciences (ISPB), 6 Avenue Rockefeller, 69008 Lyon, France; 2University Department of Psychiatry, University Hospital Centre of Saint-Etienne, 42055 Saint-Etienne, France; tristan.gabriel-segard@chu-st-etienne.fr; 3URPS-Pharmaciens Auvergne Rhône-Alpes, 194 bis Rue Garibaldi, 69003 Lyon, France; florence.durupt@urps-pharmaciens-aura.com (F.D.); asmala@wanadoo.fr (A.-S.M.); noemie.anglard@urps-pharmaciens-aura.com (N.A.); 4Public Health Service, University Hospital Centre of Saint-Etienne, 42055 Saint-Etienne, France; theophile.tiffet@chu-st-etienne.fr; 5University Department of Psychiatry, University Hospital Centre of Saint-Etienne, EA TAPE 7423, University Jean Monnet, 42055 Saint-Etienne, France; catherine.massoubre@chu-st-etienne.fr

**Keywords:** mental health, pharmacists, burnout, anxiety, depression

## Abstract

Background: The COVID-19 pandemic intensely involved pharmacists in France, with new responsibilities on a large scale, introducing to dispensary practice the performance of vaccination and nasopharyngeal swabs. This study aimed to assess the prevalence of burnout, anxiety, and depression in pharmacists after the COVID-19 health crisis and to identify factors associated with psychological distress. Methods: A cross-sectional observational study involved 1700 pharmacies in an entire French region. Sociodemographic, geographical, and medical information (burnout tested with the MBI and anxiety/depression measured on the HAD scale) were collected via an online anonymous self-administered questionnaire. The characteristics of the pharmacy and the practice of antigen testing and vaccination were requested. Quantitative and qualitative variables associated with psychological distress were investigated with a factor analysis. Results: In total, 360 responses were collected (20.5%). Of the responses, 41.9% showed definite anxiety symptoms and 18.3% showed proven depressive symptoms. Three clusters were described according to the intensity of burnout experience, depersonalization, and loss of personal accomplishment. The analysis identified that one cluster was at high risk of burnout (high burnout and depersonalization scores). Of these stressed, exhausted pharmacists, 69.3% showed definite anxiety, 37.9% showed proven depression, and in smaller pharmacies. The carrying out of antigenic testing and anti-COVID vaccination, as well as the geographical location of the pharmacy, were not discriminating factors in these three groups. Conclusion: Mental health care and suicide prevention should be provided to at-risk pharmacists. It seems essential to publicize the range of resources available to support pharmacists.

## 1. Introduction

Mental health covers a broad field, referring to individual and collective psychological equilibrium that enables individuals to maintain good health despite difficulties. As early as May 2020, the United Nations warned about concerns regarding mental health in the sanitary measures pronounced in managing the COVID-19 pandemic. Psychological distress affected many people due to the immediate health effects of the virus and the consequences of isolation, as well as the fear of being infected, dying, and losing loved ones. This sanitary crisis has highlighted population groups at risk for psychological distress: adolescents and young adults [1,2], the elderly [3], women [4], and people with pre-existing health problems [5]. It also accentuated a malaise among first-line healthcare professionals, already present before the health crisis [6,7]. The mental health of healthcare staff has given rise to several systematic literature reviews and meta-analyses focusing mainly on doctors and nurses. In 2020, Serrano-Ripoll studied the impact of viral pandemics or epidemic outbreaks (including SARS-CoV-2) on the mental health of healthcare professionals and showed increased levels of anxiety, depression, and PTSD during and after epidemics [8], with the same results in Ghahramani’s study [9] as in Pappa’s [10], who also reported that insomnia was the most frequent health problem reported. Generally speaking, the highest prevalence was found among general practitioners, nurses, and older staff. Studies in China showed more mental issues than in other countries [9]. Several reviews come to the same conclusions [11,12]. The mental health of healthcare students has also deteriorated in nursing students [13,14], dental students [15], and pharmacy students [16,17]. An observational study of 832 medical students showed signs of psychological distress in 75%, and 15% reported suicidal thoughts [18]. Burnout syndrome was defined by Freudenberger by identifying three dimensions of suffering related to work conditions: emotional exhaustion, depersonalization, and reduced personal accomplishment. It corresponds to the evaluated dimension by the Maslach scale to diagnose burnout: the Maslach burnout inventory (MBI). Exhaustion refer to an over-extended feeling, with a loss of emotional and physical energy. Depersonalization is present in an attitude and behavior of detachment from everyday elements at work and in life, notably one’s own role and performance, which corresponds to the feeling of reduced personal accomplishment, linked to a loss of hope regarding their present and future [19,20]. As front-line healthcare professionals, pharmacists have not been spared by this pandemic. As the first point of contact for patients, they are essential for dispensing prescriptions. They also played a decisive role in COVID-19 screening and vaccination. Their primary functions are advising patients, guiding them toward treatment based on risk factors, and recording validated scientific information [21,22]. The health crisis led to an increase in the number of patients, a growing shortage of medicines, and cases of harassment in the workplace with verbal abuse that clearly could impact the mental health of pharmacists [22]. A meta-analysis concerning the prevalence of mental health conditions in healthcare workers during and after a pandemic showed that the most common mental health condition was post-traumatic stress disorder, followed by anxiety disorder and major depressive disorder. Some factors were identified such as highly exposed areas of practice or geographical region and healthcare worker types (females, nurses, front-line healthcare workers exposed to patients) [23]. In France, assaults on pharmacists rose 17% in 2022 compared with 2019 [24]. Pharmacists had to modify their practices to guarantee patient care and support. They received new responsibilities by introducing to dispensary practice the performance of vaccination and nasopharyngeal swabs. While rewarding professionally, these missions generated anxiety for some or, on the contrary, a desire to move forward for others. Ishaky’s studies examined the impact of the two-year pandemic on their mental health and reported anxiety, burnout, depression, and work-related stress as a mental issue of the pandemic [25]. In addition, several individual antecedents were identified. In France, there are only two publications by Lange on the mental health of dispensing pharmacists in 2020 and 2022 in the same cohort [26,27]. Both studies used the perceived stress scale (PSS), the impact of event scale-revised (IES-R), and the Maslach burnout inventory (MBI). The same stress factors for doctors and nurses have been identified [28], including a history of psychiatric disorders, substance use disorders, and feelings of hopelessness. In France, Ghesquière highlighted the existence of risk factors, such as an increase in the number of hours worked per week, the location of the pharmacy (the risk being more significant in Nord and Pas-de-Calais), and the number of customers [29]. Several articles emphasize the protective role of resilience and mentalizing as protective mechanisms by involving metacognition processes about the self and other’s intentions and emotions, highly improving abilities to deal with stressor environments and factors, and lowering the psychological impact of the pandemic [30,31,32]. Factors that promote it were identified: social and organizational support, coping strategies, cognitive flexibility, emotional regulation, the experience of emergencies, having an optimistic, confident attitude, and specific socio-demographic data (being older, male gender, medium-high socioeconomic level) [30,31]. Regarding the recent results in the literature, the main objective of this cross-sectional survey was to assess the prevalence of burnout symptomatology and anxiety and depression symptoms in pharmacists after the COVID-19 health crisis. Therefore, we used the MBI and the HAD (hospital anxiety and depression), the same instruments as Ghesquiere in the Hauts-de-France region [29]. The second objective was to identify potential factors that may affect mental health to prevent them in the future.

## 2. Material and Methods

### 2.1. Methodology and Target Population

Our study is a descriptive study to highlight avenues for reflection and to propose a big picture about the suffering of pharmacists at work during the COVID-19 pandemic.

As of 8 February 2023, CROP (Conseil Régional de l’Ordre des Pharmaciens) counted 3033 pharmacists holding 2421 dispensaries in the Auvergne-Rhône-Alpes region. The URPS (Union Régionale des Professions de Santé) file, containing pharmacy owners only, allows questionnaires to 1700 entities to be addressed.

Four successive mailings occurred between May and June 2022, outside the COVID-19 endemic period. The URPS returned the results to all pharmacists, whether they had responded or not, at the beginning of July 2022. We obtained 360 responses from all potential pharmacies contacted, corresponding to an answer rate of 20.5%.

All participants received clear and understandable information about study objectives and procedures by mail and were free to decline participation. Informed consent was obtained from all subjects involved in the study.

The study was conducted according to the guidelines of the Declaration of Helsinki and approved by the Ethics Committee of the University Hospital Centre of Saint-Etienne (IRBN642022/CHUSTE, 19 May 2022).

### 2.2. Measures

We propose a combination of online self-questionnaires using the LimeSurvey application (Hamburg, Germany). Self-administration time was measured to approximately 10 min.

The first part of the questionnaire was related to socio-demographic and geographical data and various organizational characteristics of the dispensaries insisting on protective or aggravation factors for mental health. The pharmacists’ data concerned gender, age, length of time in the profession since graduation, lifestyle, presence of children in the household, number of weekly hours worked, stress rating in the last 7 days on a 0 to 10 scale, sleep problems, and fatigue level in the previous 7 days on a Likert scale rating 0 to 10, as well as perceived causes of work stress (feeling overwhelmed, too many patients, task interruption, administrative burden, fear of making drug dispensing mistakes, lousy atmosphere, little or no support at work, feeling of not being recognized by other professions, threat of competition, organization of COVID vaccination, antigen testing).

Pharmacy dispensary characteristics concerned geographical location (rural, semi-urban, or urban), the number of customers per day, the practice of TAG (antigenic tests) and anti-COVID vaccination, and the number of pharmacists in charge.

Psychological assessment: Our study wants to fit the one of Ghesquiere [29], allowing us to compare results between the only two French studies in the field. We evaluate burnout symptomatology to measure the suffering related to work objectively. Dimensions were assessed by the validated French version of the Maslach burnout inventory (MBI) that details three subscales: emotional exhaustion (EE), depersonalization (DP), and loss of personal accomplishment (PA) [33,34]. Burnout symptomatology is known to be positively correlated to higher scores for EE and DP dimensions. A score of the PA dimension is known to be negatively related to burnout symptoms intensity. Burnout-positive diagnosis is associated with high score levels defined by scores for EE dimension ≥ 30, for DP dimension ≥ 12, and a score ≤ 33 for PA dimension.

The prevalence of anxiety and depression symptomatology was assessed by the broadly used hospital anxiety and depression (HAD) scale, which allows the assessment of the symptomatology and infers the diagnosis of general anxiety disorder and major depressive disorder for a score ≥ 11, indicating critical symptomatology. It comprises 14 items rated by a Likert scale from 0 to 3. Seven questions are related to anxiety (total A) and seven to depression (total D). The HAD scale provides two scores (maximum score for each = 21). A score ≤ 7 is an absence of anxiety and/or depressive symptomatology. A score between 8 and 10 is considered as uncertain anxiety and/or depressive symptomatology, not sufficient to conclude an anxiety and/or depressive disorder [35,36].

### 2.3. Analysis

We began by carrying out a descriptive analysis in order to obtain an inventory of the mental health of pharmacists. Qualitative variables were expressed as number and percentage, while quantitative variables were expressed as mean, standard deviation, median, and quartiles.

We then tried to group the pharmacists according to their burnout, depersonalization, and loss of accomplishment scores, as well as self-assessed stress and fatigue levels using multiple-factor analysis and hierarchical clustering on principal components. This exploratory analysis was performed in order to see if we could identify specific factors of the pharmacists most impacted by their work. Chi-squared test was used to compare the scores across the three identified clusters according to the categorical variables (low, medium, high).

Statistical analysis was performed using R software version 4.2.1 and the FactoMineR package version 2.7.

## 3. Results

### 3.1. Descriptive Analysis (Table 1)

A.
*Characteristics of the pharmacists*


The population of respondents is 64.4% women (n = 232) and 35.6% men (n = 128). Most participants are between 40 and 55 (n = 180, 50%), 18.9% (n = 68) of participants were aged between 25 and 40, and 31.1% (n = 112) were over 55. In total, 68.1% of participants were married (n = 245), 16.7% in a civil union or cohabiting (n = 60), 8.3% divorced or separated (n = 30), 5% single (n = 18), and 1.9% widowed (n = 7). A total of 69.4% had children at home (n = 250). The majority of 59.7% (n = 215) report a dispensary activity of more than 20 years, 33.1% (n = 119) report an activity between 10 and 20 years, and 7.2% have worked fewer than 10 years since graduation (n = 26). In total, 41.9% declared to work more than 50 h a week (n = 151), 54.7% between 35 and 50 h (n = 197), and 3.3% less than 35 h (n = 12).

Regarding causes involved in work stress: 80.6% (n = 290) reported having an administrative burden; 69.4% (n = 250) had task interruptions; 51.7% (n = 186) had the impression that work deteriorated over recent years; 51.1% (n = 184) of the pharmacists felt overwhelmed; 49.4% (n = 178) thought remuneration was not representative of the work done; 45.8% (n = 165) had the feeling of not being recognized by other professions; 42.5% (n = 153) had the impression of not seeing any visibility on professional future; 30.8% (n = 111) had fear of making drug dispensing mistakes. Concerning new missions (except COVID), 44.4% (n = 160) felt involved, 36.9% (n = 133) were a distant concern, 12.5% (n = 45) felt stressed, and 6.11% n = 222) reported feeling not concerned.

B.
*Characteristics of the pharmacies*


The location of the dispensary is essential, shaping the population of patients. In total, 30.4% (n = 131) of pharmacies were located in rural areas, 33.9% (n = 122) in semi-urban areas, and 29.7% (n = 107) in urban areas. Daily number of patients is a determinant factor for psychological burden: 15.6% (n = 56) of pharmacies received fewer than 100 patients per day, 43.6% (n = 157) 100 to 200 patients per day, 33.6% (n = 121) 200 to 400 patients per day, and 7.2% (n = 26) more than 400 per day. Pharmacists were alone in the dispensary for 14.4% (n = 52) of respondents, 2 to 3 for 61.7% (n = 222) of respondents, and 4 or more for 23.9% (n = 86) of respondents. Regarding practices specific to the pandemic, 76.9% of dispensaries practiced antigen tests, and 92.8% practiced COVID-19 vaccination.
ijerph-20-06988-t001_Table 1Table 1Characteristics of the population and the pharmacy.Characteristics of the PopulationItemsN (%)GenderFemale232 (64.4%)Male128 (35.6%)Age25 to 40 years68 (18.9%)40 to 55 years180 (50%)Over 55 years112 (31.1%)LifestyleMarried245 (68.1%)Cohabiting/PACS60 (16.7%)Divorced/separated30 (8.3%)Single18 (5%)Widowed7 (1.9%)The presence of children at homeNo110 (30.6%)Yes250 (69.4%)Length of time in the profession(since graduation)Under 10 years26 (7.2%)10 to 20 years119 (33.1%)Over 20 years215 (59.7%)Number of hours worked per weekUnder 35 h12 (3.3%)35 to 50 h197 (54.7%)Over 50 h151 (41.9%)**Characteristics of the pharmacy**

Pharmacy typologyRural131 (36.4%)Semi-urban 122 (33.9%)Urban 107 (29.7%)Number of customers per day in the pharmacy0 to 10056 (15.6%)100 to 200157 (43.6%)200 to 400121 (33.6%)Over 40026 (7.2%)Number of pharmacists in the pharmacy152 (14.4%)2 to 3222 (61.7%)4 and more86 (23.9%)Antigen tests practiced in the pharmacyNo83 (23.1%)Yes277 (76.9%)COVID vaccination in the pharmacyNo26 (7.2%)Yes334 (92.8%)


### 3.2. Clinical Characteristics of the Population (Table 2)

The self-evaluation of stress and fatigue levels over the last 7 days was performed using a Likert scale (rating from 0 to 10). The stress mean score was 5.73 (SD = 2.6), and the fatigue mean score was 6.24 (SD = 2.5).A.*Mental health scales assessment:*

The MBI scale assessment revealed a high emotional exhaustion (EE) level of 44.2% (n = 159), 25.3% (n = 91) presented a score considered as a moderate level, and 30.6% (n = 110) were regarded as a low level of burnout symptomatology. Regarding depersonalization (DP) scores, 39.7% (n = 143) of pharmacists reported scores corresponding to a high level of depersonalization, 27.8% (n = 100) had scores corresponding to a moderate level, and 32.5% (n = 117) had scores corresponding to a low level. The last dimension of the MBI scale, loss of personal accomplishment (PA), reported an important level for 15.8% (n = 57) of participants, and 29.2% (n = 105) of participants reported a score corresponding to a moderate level. A total of 55% (n = 198) of the pharmacists reported a score corresponding to a low level.

The HAD scale reported a score corresponding to specific symptomatology for anxiety disorder of 41.9% (n = 151). A total of 36.7% (n = 132) and 36.7% (n = 132) of participants did not report anxiety symptomatology. In total, 18.3% (n = 66) of participants said specific symptomatology of depressive disorder, 18.3% (n = 66) of participants presented uncertain depressive symptomatology, and 63.3% had no symptomatology (n = 228).

B.
*Identification and description of pharmacist clusters (Appendix A)*


Based on these findings, the clustered analysis identified three clusters of pharmacists on the levels of emotional exhaustion (EE), depersonalization (DP), and loss of personal accomplishment (PA) to study characteristics that are the most represented in each subpopulation. Graphic representations of clusters’ comparison by the chi^2^ test are available in Appendix A.
-Cluster 1: 28.9% (n = 104)

Regarding burnout symptoms, the vast majority reported a low level of EE (97.12%, n = 101), 2 (1.92%) reported a high level of EE, and 1 (0.96%) had a moderate level. Regarding DP, 60.6% of participants (n = 63) had a low level, 23.1% (n = 24) had a moderate level, and 16.3% (n = 17) reported a high level of DP. Regarding PA, the low-level concern was 82.7% (n = 86), moderate level for 13.5% (n = 14), and high level for 3.85% (n = 4). The average score for anxiety was 5.78 (SD 2.85). Definite anxious symptoms concerned 5.77% (n = 6) of participants, 22.1% (n = 23) had doubtful symptoms, and 72.1% (n = 75) did not present anxiety symptoms. The average score for depression was 2.62 (SD 1.94). No pharmacist reported definite depressive symptomatology, only 2.8% (n = 3) doubtful symptomatology, and 97.1% (n = 101) had no depressive symptomatology. The average score for stress was 3.15 (SD = 2.14), and the average score for fatigue was 3.65 (SD = 2.4). 70.2% of the pharmacists in this cluster worked between 35 and 50 h a week (n = 73).
-Cluster 2: 25% (n = 90)

Regarding burnout symptoms, all the pharmacists of cluster 2 reported a moderate level of EE. 31.1% (n = 28) of participants reported a high level of DP, 32.2% (n = 29) reported a moderate level, and 36.7% (n = 33) had a low level of DP. In total, 11.11% (n = 10) reported a high level of PA, 31.1% (n = 28) moderate level, and 57.8% reported a low level of PA (n = 52). The average score for anxiety was 8.63 (SD 3.31). In total, 33.33% (n = 30) presented definite anxious symptoms, 25.6% (n = 23) had doubtful symptoms, and 41.1% (n = 37) did not show anxiety symptoms. The average score for depression was 5.6 (SD 2.71). Three pharmacists reported definite depressive symptomatology (3.33%), 21.11% (n = 19) doubtful symptomatology, and 75.6% (n = 68) did not report depressive symptomatology. The average score for stress was 5.48 (SD 1.85), and the average score for fatigue was 6.59 (SD 1.66).
-Cluster 3: 46.1% (n = 166).

Regarding burnout symptoms, 94.58% (n = 157) reported high levels of EE, and the others, 5.42% (n = 9), presented low levels of EE. A total of 59% (n = 98) of participants reported a high level of DP, 28.3% (n = 47) moderate level, and 12.7% (n = 21) low level of DP. In total, 38% of participants (n = 63) reported a moderate level, and 36.1% (n = 60) reported a low level of PA. 25.9% (n = 43) reported a high level of PA. The average score for anxiety was 12.6 (SD 3.96). A total of 69.28% of participants (n = 115) had definite anxious symptoms, 18.7% (n = 31) had doubtful symptoms, and 12% (n = 20) did not report anxiety symptoms. The average score for depression was 9.4 (SD 3.61). In total, 37.95% (n = 63) of pharmacists reported definite depressive symptomatology, 26.51% (n = 44) presented doubtful symptomatology, and 35.5% (n = 59) did not report depressive symptomatology. The average score for stress was 7.48 (SD 1.66), and the average score for fatigue was 7.67 (SD 1.49).

A closer look at this group revealed a higher level of stress and fatigue than cluster 2, with many of them living alone. No pharmacy typology stood out about location. These pharmacists often worked more than 50 h a week and were older.
ijerph-20-06988-t002_Table 2Table 2Clinical characteristics of the population.Stress level over the past 7 days on a scale of 0 to 10Average (SD)5.73 (2.6)Median [Q1; Q3]6 [4; 8]Fatigue level over the past 7 days on a scale of 0 to 10Average (SD)6.24 (2.5)Median [Q1; Q3]7 [5; 8]**MBI**

n (%)High159 (44.2%)Moderate 91 (25.3%)Low110 (30.6%)Depersonalization (DP)High143 (39.7%)Moderate 100 (27.8%)Low117 (32.5%)Loss of personal accomplishment (PA)
High57 (15.8%)Moderate 105 (29.2%Low198 (55%)**HAD**

HAD: anxietyDefinite symptomatology 151 (41.9%)Doubtful132 (36.7%)symptomatology 132 (36.7%)No symptomatology
HAD: depressionDefinite symptomatology 66 (18.3%)Doubtful66 (18.3%)symptomatology 
No symptomatology228 (63.3%)**Clusters**
123n (%)n (%)n (%)Emotional exhaustionHigh2 (1.92%)0 (0%)157 (94.58%) Moderate1 (0.96%)90 (100%)0 (0%)Low101 (97.12%)0 (0%)9 (5.42%)DepersonalizationHigh17 (16.3%)28 (31.1%)98 (59%)Moderate24 (23.1%)29 (32.2%)47 (28.3%)Low63 (60.6%)33 (36.7%)21 (12.7%)Loss of personal accomplishmentHigh4 (3.85%)10 (11.11%)43 (25.9%)Moderate14 (13.5%)28 (31.1%)63 (38%)Low86 (82.7%)52 (57.8%)60 (36.1%)HAD: anxiety score Average (SD)5.78 (2.85)8.63 (3.31) 12.6 (3.96)Median [Q1; Q3]5 [3.75; 8] 8 [6; 11] 13 [10; 16] HAD: depression score Average (SD)2.62 (1.94)5.6 (2.71)9.4 (3.61)Median [Q1; Q3]2 [1; 4]5 [4; 7]9 [7; 12]HAD: anxietyDefinite symptomatology 6 (5.77%)30 (33.33%)115 (69.28%)Doubtful symptomatology 23 (22.1%)23 (25.6%)31 (18.7%)No symptomatology75 (72.1%)37 (41.1%)20 (12.0%)HAD: depressionDefinite symptomatology 0 (0%)3 (3.33%)63 (37.95%)Doubtful symptomatology 3 (2.8%)19 (21.11%)44 (26.51%)No symptomatology101 (97.1%)68 (75.6%)59 (35.5%)Stress level over the past 7 days on a scale of 0 to 10Average (SD)3.15 (2.14)5.48 (1.85)7.48 (1.66)Median [Q1; Q3]3 [2; 5]6 [4; 7]8 [7; 8.75]Fatigue level over the past 7 days on a scale of 0 to 10Average (SD)3.65 (2.4)6.59 (1.66)7.67 (1.49)Median [Q1; Q3]3 [2; 5.25]7 [5; 7.75]8 [7; 9]


### 3.3. Factors Associated with Burnout Symptoms:

Having a partner did not make it possible to work less, and there was no protection against burnout and stress. Working alone, on the other hand, encouraged burnout and stress. Working more than 50 h a week was a risk factor for burnout. This study did not show the relationship between the size of the pharmacy and the level of suffering experienced by the owner. Living with a partner and having children at home did not seem to influence suffering at work negatively, nor did it appear to be a protective factor.

## 4. Discussion

Our is the first study on the mental health of pharmacists in the Auvergne Rhône-Alpes region, home to 12% of the French population. This survey focuses on dispensing pharmacists’ mental state after an unprecedented situation that has turned working habits and skills upside down. In France, very few studies were published about pharmacist’s health. Only two studies concerning the same cohort regarding the impact of the COVID-19 epidemic on mental health were conducted in 2020 [23,24].

### 4.1. The Mental Health of Pharmacists Needs Concern

Most of the studies are cross-sectional studies. Some reviews and meta-analyses recently highlighted psychological and psychiatric consequences on pharmacists’ health.

Burnout is not currently defined as a psychiatric disorder, representing a mental health issue. The MBI was chosen as the instrument of choice, recognized for its specificity, and validated in French. Despite the absence of a global score for MBI, the three sub-scores can help describe the suffering of professionals. The first and most central dimension is emotional, psychic, and physical exhaustion (feeling emptied of resources). The second dimension, depersonalization, corresponds in a way to a movement of self-preservation in the face of the (emotional) demands of the profession that the person can no longer face. They have decreased personal accomplishment at work. In its third dimension, burnout is characterized by a loss of personal achievement and a devaluation of oneself, reflecting both for the individual the feeling of being ineffective in his work and not being up to the job. Limitations in interpreting MBI results are linked to the absence of an overall score and the need to work on three sub-scores. Maslach justifies this by pointing out that burnout is multidimensional. There is a moderate correlation between emotional exhaustion and depersonalization scales. Furthermore, the loss of personal accomplishment scale is independent of the other two. However, we can still consider that a loss of personal accomplishment is a consequence of the two dimensions. Our study revealed a high level of emotional exhaustion (44.2% of the pharmacists), depersonalization (39.7%), and loss of personal accomplishment (15.8%). The group of pharmacists, comprising cluster 3, presented considerable mental health characteristics according to high suffering compatible with burnout syndrome. Half of the studies (n = 14) in Ishaky’s review examined burnout among pharmacists, and six studies used the MBI [25]. Results of burnout severity and prevalence were inconsistent, depending on the assessment period [25]. In Lebanon, the prevalence of moderate-to-high personal and work-related, and client-related burnout was 77.8%, 76.8, and 89.7%, respectively [32]. A French study conducted for a pharmacy thesis by Ghesquière, using MBI, reported that 37.4% showed severe emotional exhaustion, 32.7% high depersonalization, and 23.4% low loss of personal accomplishment [29]. Indeed, the COVID-19 pandemic burden also affects the pharmacists in our study, leading to burnout symptoms.

Burnout syndrome is strongly associated with comorbidities (anxiety, depression, fatigue, work-related stress, and sleep disorders) and frequent use of anxiolytics. Our study found 41.9% of pharmacists with anxiety disorder and 36.7% with doubtful symptomatology. A total of 18.3% reported a depression, and 18.3% doubtful symptomatology. Worryingly, 69.28% of cluster 3 had definite anxious symptomatology, and 37.95% had definite depressive symptomatology. These results reveal a high level of psychological distress among senior pharmacists. We did not, however, investigate psychiatric histories or the use of psychotropic drugs in the study population, although several studies have shown a link between burnout and psychopathology [37,38]. Anxiety and depression syndromes linked to the pandemic experience were studied and revealed highly worrying results. Dhindayal’s study assessed 953 South African pharmacists, 44.5% of whom worked in dispensaries, and they showed 66.1% anxiety symptoms, 62.9% depression symptoms, and 73.8% stress [39]. These resulted in a low quality of life for a considerable proportion of 51.5% of them. In a sample of over 1200 nurses and doctors working in China, 50% reported symptoms of depression, 45% anxiety, 34% insomnia, and 71% distress. Among their medical staff, 36% had insomnia symptoms [40]. Lange’s initial findings regarding the impact of the COVID-19 epidemic on mental health also indicated significant psychological disorders among healthcare professionals [27], including post-traumatic stress disorder in 23.1% of them during lockdown, then 16.4% two years later [26]. In the French study by Ghesquiere, 17.9% (n = 37) of the population used anxiolytics. This subgroup presented anxiety disorders in 97.2% of cases and depression symptomatology scores in 40.5% [29].

### 4.2. Pharmacist Mental Health Impairment Is Linked to Risk Factors

In our study, the risk factors identified were the number of hours worked per week (more than 50 h), working alone, and living alone. The pharmacists from cluster 3, with the lowest mental health condition, were also older than other clusters. Pharmacists from clusters 2 and 3 reported higher fatigue levels than cluster 1. These results could be interpreted as a factor associated with professional exhaustion or as a first symptom of burnout. These findings are consistent with those previously described in the French study. Ghesquière highlighted the existence of risk factors, such as an increase in the number of hours worked per week, the location of the pharmacy (the risk being more significant in Nord and Pas-de-Calais locality), and the number of customers per day [29]. Conversely, hiring an additional full-time assistant pharmacist was not perceived as a protective factor and was even a moderate risk factor for burnout linked to loss of personal accomplishment. In addition, pharmacists at risk of high burnout scores were full pharmacists, especially those with seniority [29]. Our results are consistent with those from the Dhakal et al. study, reporting that 77.9% of pharmacists were severely disturbed. Aggravating factors identified were daily working hours (8 h or more), job dissatisfaction, and experience of over 3 years [41]. The pandemic caused an increase in weekly hours and a decrease in days off per month.

Buomprisco’s study reported on a large sample of 401 participants (dispensing and hospital pharmacists) that older pharmacists and those with more experience reported more psychological stress. This author identified that women who felt lonely at home reported higher psychological stress [42]. Dhindayal et al. showed that significant risk factors for poor mental health included being female, living apart from family during the COVID-19 pandemic, and pre-existing poor mental health [39].

Youssef et al. identified that younger age, being a staff pharmacist, working more than 40 h per week, and a high perception of the COVID-19 threat was associated with a moderate-to-high likelihood of burnout in all three domains [43]. According to Piro, emotional exhaustion tended to be higher among women, even though their working hours were shorter when employed [44]. It was the inequitable distribution of domestic work, however, that made their daily lives more difficult and stressful [45,46]. Others pointed to marital status: burnout syndrome would affect people with a partner and children less. Women were more likely than men (19%) to talk about their professional worries to someone close to them (56%) [44]. In our study, these factors did not emerge as significant.

Our study identified some factors as stressors: feeling overwhelmed, experimenting with task interruptions, and burdened by too many administrative tasks. These findings are according to those from the other French survey. The most frequently cited stress factors were increasingly demanding, even violent and morally distressed patients, little visibility of the professional future, frequent interruptions in work, and the impression of having too many administrative burdens [29]. Several studies have identified other stressors. Pharmacists have major responsibilities, such as advising, informing, and educating patients. Burnout is often linked to patient trust in pharmaceutical care [15]. Coelho’s article reports that 44.4% (of the 250 participants) felt that their health was slightly worse after the start of the pandemic. The highest levels of exhaustion were related to patient relations [47]. Pharmacists were sometimes the only ones to provide patients with protective devices, especially at the pandemic’s start. For example, the lack of surgical masks may have led to stress in the face of pressing patient demands [48]. In our study, the major concerns of pharmacists were having a fear of making drug dispensing mistakes, having the impression that work deteriorated over recent years, thinking remuneration was not representative of the job done, and having the appearance of not seeing any visibility in the professional future. Pharmacists were also affected by the irregular flow of patients during the day. In 2007, 55% of pharmacists felt that their work had deteriorated. Paradoxically, however, 90% felt useful, 73% were proud of a well-done job, and 52% felt recognized for their work [44]. Lange insists on human resources management: management, management of employment contracts, the 35-h working week, RTT (reduced working time) and paid leave, unscheduled absences, maternity leave, staff moods, and conflicts within the team [26]. Declining profit margins and pressure from banks could also lead to significant financial stress [29].

### 4.3. Proposition to Increase Resilience and Lower the Mental Health Impact

In the pandemic management, it appears crucial to implement preventive measures and psychological support to alleviate pharmacists’ distress, as proposed in other studies [41,42]. Interventions at personal and national levels are needed to increase pharmacists’ well-being by reducing stress, improving self-efficacy and resilience, and preventing burnout. It first appears essential to publicize the range of resources available to support pharmacists, with assistance also going toward telemedicine and self-care strategies. Pharmacists who had received training in COVID-19 management were more confident and effective. Prevention strategies such as work-life balance, peer support, continuing education, and self-care effectively reduced the risk of burnout [15]. Mentalizing training also appears as an efficient strategy to protect against burnout evolution in increasing the ability to understand self and other’s intentional mental states, leading to reinforced resilience and more efficient adaptation to stressors [31].

Four domains of intervention are proposed as a strategy to increase resilience and limit the psychological impact of stressors as pandemic management. Recommendations regarding healthcare workers were submitted by Halms et al. after the synthesis of 41 studies [49]. Four domains were identified: social and structural support, grouping the need for healthcare workers of appreciation from the hierarchy or general public; the demand for social support that could be provided by family, friends, partners, or coworkers; recommendation of staff retention consisting in well human management; and proposition of everyday support to offer better well-being to healthcare worker in supporting daily tasks. The second category corresponds to the work environment. This is grouping better working conditions that include adapted personal protective equipment such as face masks and training of supervisors to promote professional development and work well-being (also recommended in a study of nurse quality of life during the pandemic [50] and best practice protocols that ensure the safety of clinical procedures. The third domain is communication, grouping recommendations to use news and social media sources of information, and improving reliable information transmission between healthcare workers. The last domain is mental health support. This recommends proposing help hotlines to dispense mental support while maintaining anonymity; early identification of high-risk individuals for mental health issues; some recommendations also consist in access to mental health services; developing self-care is broadly recommended in promoting peer support by self-help groups, team cohesion initiatives, guidance on resilience, stress management strategy formation [49].

From the results of our study on the high prevalence of burnout, we need to look at preventive solutions at the organizational level of pharmacists’ orders. The hotline support, such as ADOP in the Rhône-Alpes region or the national number 3114, is specialized to support suicidal intentions and is broadly reminded to all pharmacies in the area. Improvement of management methods to avoid burnout among pharmacists and their staff is currently underway. Good hygiene and optimizing daily activities distribution were broadly encouraging to lead to a better-balanced life, with areas of involvement other than work. Resilience needs to be enabled at the collective and individual levels by using social and family support, recreational activity, and promoting informative sources about mental health and psychological resources at the disposal of healthcare workers [51].

Our study suffers from several limits. First, the participation rate is 20.5%, which does not broadly represent the whole dispensary’s characteristics and the study’s observational design. We also could invocate a bias in responses that were more provided by pharmacists who experienced mental health issues during the COVID-19 pandemic.

## 5. Conclusions

The study revealed that almost half of all dispensing pharmacists in the Auvergne-Rhône-Alpes region suffered psychological distress. Even if this profession was not very optimistic about its future, the HPST Law (Hospital, Patients, Health and Territory) has contributed significantly to the dynamism of the pharmacy. Introducing new missions (such as therapeutic interviews and vaccination) has placed greater demands on pharmacists and enhanced their role as health professionals.

The COVID-19 health crisis demonstrated the importance of the French network of dispensing pharmacies in its regular presence throughout the country and its unfailing response to the workload.

## Data Availability

URPS-Pharmaciens Auvergne Rhône-Alpes (Lyon) and University Hospital Centre of Saint-Etienne, France.

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
