# Peer review of "Survey on the Mental Health of Dispensing Pharmacists in the Auvergne-Rhône-Alpes Region (France)"

_ijerph, 2023, doi:10.3390/ijerph20216988_

Round 1

Reviewer 1 Report (Previous Reviewer 1)

Comments and Suggestions for Authors

Dear Authors,

The manuscript has been improved, and I recommend that the paper be accepted without any changes and published in the journal IJERPH.

Author Response

This reviewer does not require any modifications to the article

Reviewer 2 Report (Previous Reviewer 2)

Comments and Suggestions for Authors

Dear authors, thank you for your work. I believe tha some more improvement could be brought.

Introduction - this section must be improved with paragraph about bournout and an overview about the factors that may affect mental health and why the identify these factors.

Analysis and results - I think ANOVA or MANOVA should be added to compare the scores of the 3 clusters identified. It would be not just interesting but also useful in terms of giving more consistency to the discussion.

Author Response

To Reviewer 2:

Thanks for your comments to improve our manuscript. We took into account your proposition to add a paragraph in the introduction. We added burn out information (lines 58-66), and about impacting factors in mental health (lines 73 to 78)

About the statistic analysis: it is indeed interesting and useful to compare the scores across the three identified clusters. However, as the scores are categorical variables (low, medium, high) we used chi squared test instead of Anova. A sentence has been added lines 171-173, and lines 238-239. And figures are available as supplemental.

Reviewer 3 Report (Previous Reviewer 3)

Comments and Suggestions for Authors

I want to thank you for estimating the suggestions provided for the manuscript, and that these have been considered. Congratulations on your manuscript

Author Response

This reviewer does not require any modifications to the article

This manuscript is a resubmission of an earlier submission. The following is a list of the peer review reports and author responses from that submission.

Round 1

Reviewer 1 Report

Comments and Suggestions for Authors

Dear Authors,

I appreciate the opportunity to review the article titled "Survey on the mental health of dispensing pharmacists in the Auvergne-Rhône-Alpes region (France)". The current study has no flaws in ethics, trial design, methods, or statistics. The study appears to follow relevant guidelines and provides an original contribution to the existing scientific literature. There are no flaws in the data presented, and there are no misleading or false conclusions.

I have to say that this is a very well-written manuscript, and it is a nice and original study of this topic. The manuscript deals with psychological distress in pharmacists after the COVID-19 health crisis, a topic that continues to be considered interesting by professionals and students from different fields.

However, before proceeding with its publication, some aspects must be revised: 

1)   The abstract must be improved. The abstract must begin by placing the question being discussed in a broad context and then outline the purpose of the study.

2)   When it comes to the practical implications of the study, in addition to the aforementioned preventive strategies such as work-life balance, receiving training in COVID, peer support, continuous education, and self-care that are effective in reducing the risk of burnout, I strongly suggest the authors include in the practical implications the latest scientific findings that mentalizing as well as resilience are key to mental health because the studies that were conducted on a sample of healthcare workers during the COVID-19 pandemic revealed that good capacity for mentalizing and resilience can reduce burnout and psychological distress.

3)   I suggest to the authors that after this sentence "Interventions at personal and national levels are needed to increase pharmacists well-being by reducing stress, improving self-efficacy and resilience, and preventing burnout.", they add an adequate reference as well as an important finding related to encouraging and strengthening the mentalizing capacity of all employees in the healthcare sector, including pharmacists.

I recommend the article for publication in the International Journal of Environmental Research and Public Health (IJERPH) after minor revisions. I wish you all the best as you work on the revision of this paper. 

Sincerely,

Reviewer

Author Response

Reviewer 1: 

1) The abstract must be improved. The abstract must begin by placing the question being discussed in a broad context and then outline the purpose of the study. 

Thanks for your comment; we modified the abstract according to your recommendations to improve the comprehension. 

2) When it comes to the practical implications of the study, in addition to the aforementioned preventive strategies such as work-life balance, receiving training in COVID, peer support, continuous education, and self-care that are effective in reducing the risk of burnout, I strongly suggest the authors include in the practical implications the latest scientific findings that mentalizing as well as resilience are key to mental health because the studies that were conducted on a sample of healthcare workers during the COVID-19 pandemic revealed that good capacity for mentalizing and resilience can reduce burnout and psychological distress. 

Thank you for the remark. We added a part about resilience and mentalizing protective factors. 

3) I suggest to the authors that after this sentence "Interventions at personal and national levels are needed to increase pharmacists well-being by reducing stress, improving self-efficacy and resilience, and preventing burnout.", they add an adequate reference as well as an important finding related to encouraging and strengthening the mentalizing capacity of all employees in the healthcare sector, including pharmacists. 

Thank you for this proposition. We added an encouragement to mentalizing training and intervention following results of Safiye et al. 2023.  

Reviewer 2 Report

Comments and Suggestions for Authors

First of all, I want to thank you the authors for their contribution which could be helpful to understand not just the direct implication related to Covid-19 but also the impact on the long term for healthcare professionals. Below some indication to improve the quality of the paper

The introduction must be improved and integrated with an analysis of malaise at the workplace.

The Material and Methods section must be improved, providing further explanation of the process (i.e. the reasons for choosing those specific scales)

The discussion should be organised with sections to provide a better explanation of the results and the connections with the theory.

Comments on the Quality of English Language

Moderate editing of English language required

Author Response

Reviewer 2: 

The introduction must be improved and integrated with an analysis of malaise at the workplace. 

Thank you for your remark, we enriched the introduction with lots of references and a focus on the mental health issues encountered during COVID-19 pandemic in health department from line 

The Material and Methods section must be improved, providing further explanation of the process (i.e. the reasons for choosing those specific scales) 

We took into account your welcome comments and precised criteria that drove us to choose the scales (MBI and HAD). 

The discussion should be organised with sections to provide a better explanation of the results and the connections with the theory. 

Thanks for your advice; we redesigned the discussion to improve the clarity of results with sections and more confrontation to the literature.

Reviewer 3 Report

Comments and Suggestions for Authors

The authors of the manuscript handle data that can be treated in a very interesting way, and carry out research that provides conclusions of interest to the scientific community. However, the presentation and writing of the manuscript presents poor structuring, writing, presentation of results (including tables) and, in general, a poorly written manuscript to be able to be submitted to the journal. The manuscript needs a deep writing to be able to be published.

Author Response

Reviewer 3:  

The authors of the manuscript handle data that can be treated in a very interesting way, and carry out research that provides conclusions of interest to the scientific community. However, the presentation and writing of the manuscript presents poor structuring, writing, presentation of results (including tables) and, in general, a poorly written manuscript to be able to be submitted to the journal. The manuscript needs a deep writing to be able to be published. 

We took into account your comment and deeply modified the introduction, results section and discussion. Some of the data in the tables have been included in the appendix. We would have appreciated further precise and constructive remarks.

            A native English-speaking editor has proofread the manuscript; all authors have approved it and agree with its submission to your journal.

            Thank you for your thoughtful consideration of our submitted paper. We hope that you find our materials in good order and look forward to hearing from you at your earliest convenience regarding the paper’s disposition for publication.